# Integrated care pathways in neurosurgery: A systematic review

Keng Siang Lee[1,2]*, Stefan Yordanov[2], Daniel Stubbs[3], Ellie Edlmann[2], Alexis Joannides[2], Benjamin Davies[2]*

1 Bristol Medical School, Faculty of Health Sciences, University of Bristol, Bristol, United Kingdom, 2 Division of Neurosurgery, Department of Clinical Neurosciences, University of Cambridge, Cambridge, United Kingdom, 3 Division of Anaesthesia, Department of Medicine, Addenbrooke's Hospital, University of Cambridge, Cambridge, United Kingdom

* kl17746@bristol.ac.uk (KSL); bd375@cam.ac.uk (BD)

**Data Availability Statement:** All relevant data are within the manuscript and its Supporting Information files.

**Funding:** The author(s) received no specific funding for this work.

## Abstract

### Introduction

Integrated care pathways (ICPs) are a pre-defined framework of evidence based, multidisciplinary practice for specific patients. They have the potential to enhance continuity of care, patient safety, patient satisfaction, efficiency gains, teamwork and staff education. In order to inform the development of neurosurgical ICPs in the future, we performed a systematic review to aggregate examples of neurosurgical ICP, to consider their impact and design features that may be associated with their success.

### Methods

Electronic databases MEDLINE, EMBASE, and CENTRAL were searched for relevant literature published from date of inception to July 2020. Primary studies reporting details of neurosurgical ICPs, across all pathologies and age groups were eligible for inclusion. Patient outcomes in each case were also recorded.

### Results

Twenty-four studies were included in our final dataset, from the United States, United Kingdom, Italy, China, Korea, France, Netherlands and Switzerland, and a number of sub-specialties. 3 for cerebrospinal fluid diversion, 1 functional, 2 neurovascular, 1 neuro-oncology, 2 paediatric, 2 skull base, 10 spine, 1 for trauma, 2 miscellaneous (other craniotomies). All were single centre studies with no regional or national examples. Thirteen were cohort studies while 11 were case series which lacked a control group. Effectiveness was typically evaluated using hospital or professional performance metrics, such as length of stay (n = 11, 45.8%) or adverse events (n = 17, 70.8%) including readmission, surgical complications and mortality. Patient reported outcomes, including satisfaction, were evaluated infrequently (n = 3, 12.5%). All studies reported a positive impact. No study reported how the design of the ICP was informed by published literature or other methods

**Competing interests:** The authors have declared that no competing interests exist.

## Conclusions

ICPs have been successfully developed across numerous neurosurgical sub-specialities. However, there is often a lack of clarity over their design and weaknesses in their evaluation, including an underrepresentation of the patient's perspective.

## Introduction

Integrated care pathways (ICPs) are a pre-defined framework of evidence based, multidisciplinary practice for specific patients. They aim to ensure patients move more effectively through a clinical experience [1]. lCPs outline care processes for the management of a specific condition [2]. As such they can define the patient's journey through a bounded health system and may, in certain disciplines, transcend organisational boundaries. Their overall aim is to ensure delivery of timely and efficient care to maximise patient outcomes. ICP can thus seamlessly integrate evidence-based practice into day-to-day care, whilst providing a framework for ongoing clinical audit. Process steps within an ICP may be incorporated for other reasons including to mitigate points of system risk and ensure continuity of care, patient safety and satisfaction, efficiency gains, teamwork, and staff education [3–5].

ICPs are therefore best suited, to well defined patient populations with common and consistent care requirements [6, 7]. A well-known example driven by clear national guidance is the care framework for patients with a fractured neck of femur. Local adoption of such processes has been shown to reduce morbidity and mortality as well as hospital length of stay [8, 9]. Due to the success of such initiatives, national reporting infrastructures such as the National Hip Fracture Database (NHFD) have been created to enable ongoing audit and facilitate payment of a best-practice-tariff [10]. Despite the impact of this framework its development was not reached in a systematic way but an increasing body of literature advocates for the coordinated design and engineering of healthcare systems in order to minimise risk and improve outcomes, with such a 'systems approach' endorsed by various medical royal colleges [11–13].

## Aims and objectives

The overarching aim of this study is to identify how ICPs have been employed in neurosurgery for patients of any age. In so doing we will identify:

- The areas of neurosurgical practice where ICPs have been adopted

- To identify the impact of ICPs in published studies and the criteria by which this is judged

- Identify how ICPs were developed

- Identify common themes across ICPs that may be related to successful ICP adoption.

To the best of our knowledge this is the first systematic review to aggregate published data on ICPs for neurosurgical diseases.

## Methods

This review was conducted in accordance to the Preferred Reporting Items for Systematic Reviews and Meta-Analyses (PRISMA) guidelines (S1 Checklist) [14]. PROSPERO registration was obtained (registration number CRD42020199650).

## Search strategy

A search string was developed to identify original research studies reporting ICPs in neurosurgery (see S1 Table). The following databases were searched on the 20ᵗʰ July 2020: Ovid Medline, Embase and Cochrane Central Register of Controlled Trials (CENTRAL).

## Definition of ICP

The medical literature is inconsistent over the core or minimum features of an ICP. Multiple synonyms–clinical pathways, critical pathways, care maps, care protocols, and multidisciplinary plans–have been used [15]. One review identified 84 different definitions for ICPs in a Medline search between 2000 and 2003 [15].

For the purposes of this systematic review, an ICP was defined in accordance with the definition developed by the European Pathway Association (EPA) [6]. This definition has been used in other reviews of ICPs [16, 17]. From this definition, key characteristics of an ICP must include:

1. An explicit statement of the goals and key elements of care based on evidence, best practice and patient expectations.

And should include:

2. The facilitation of communication, coordination of roles, and sequencing of activities of the multidisciplinary care team, patients and their relatives.

Whilst enabling:

3. The documentation, monitoring, and evaluation of variances and outcomes.

4. The identification of the appropriate resources.

## Study selection

All titles and abstracts were independently screened by two reviewers (KSL and SY) against a set of pre-defined eligibility criteria (S2 Table). Potentially eligible studies were selected for full-text analysis. Disagreements were resolved by consensus or appeal to a third senior reviewer (BD). Agreement among the reviewers on study inclusion was evaluated using Cohen's kappa [18].

All original studies reporting the details of the ICPs and outcomes of patients with any neurosurgical disease were included in our systematic review. Case series were included. Studies of small sample sizes were included per recommendations by the Cochrane Statistical Methods Group and in accordance with methodologies of previously published meta-analyses [19–21]. Other exclusion criteria included non-English articles, non-original research papers, laboratory-based and epidemiological studies, and non-human research subjects as these were deemed to not provide relevant information needed in this paper (see S2 Table). If data from the same patient population was published several times or overlaps in more than one article from the same institution, the publication that reported the largest sample size data was selected.

**Risk of bias assessment.** The quality of included studies was assessed using the Joanna Briggs Institute (JBI) checklist for non-randomised experimental studies [22]. Full details are in S3 Table (S3 Table). In summary, these tools rated the quality of selection, measurement and comparability for all studies and gave a score for experimental studies (maximum of 9)

and case series (maximum of 10). Two researchers (KSL and SY) assessed the quality of all included studies and discussed discrepancies until consensus was reached.

### Data extraction

Data were extracted on the following variables: study details, sample size, patient demographics, type of neurosurgical diseases ICP was used for, components of ICP, outcome measures, factors for success/failure of ICP.

### Statistical analysis

Data were organised and tabulated to allow inspection and investigation of patterns within the data. Given their heterogeneity, formal meta-analysis of studies was not possible. Data is therefore reported narratively, with descriptive statistics only.

## Results

### Characteristics and quality of included studies

Number of articles screened and selected for inclusion are shown in Fig 1. Using the designated search terms, a total of 1769 unique articles were identified and 24 were included in the final dataset [23–46]. Reliability of study selection between observers was substantial at both the title and abstract screening stage (Cohen's κ = 0.79) and the full-text review stage (Cohen's κ = 0.87) [18].

The characteristics of the studies are shown in Table 1. Thirteen studies were from the United States (US), five from the United Kingdom (UK), and one each from China, Korea, Italy France, Netherlands and Switzerland. Sub-specialities represented in these studies included skull base, neurovascular, neuro-oncology, and spinal neurosurgery. These are shown graphically in Fig 2.

All studies were non-randomised. We included 13 cohort studies with a control group and 11 case series without a control group. All 13 cohort studies [23–35], attained a score of 9 out of 11 on the JBI checklist for cohort studies (see S3 Table), whilst 10 of the 11 case series [36–46], attained a full score of 10 on the JBI checklist for case series, with one study scoring 8 (S4 Table). We observed that the majority of the primary studies (18 of 24 (75%)) included were published from year 2010 onwards. Fig 3 illustrates this trend, with a rise in cohort studies (comparator design) and case series (no comparator) published.

### Elements of ICP

Eligible participants for inclusion in this systematic review were patients in secondary and tertiary care settings which includes the coordination and continuity of healthcare as patients transfer between different locations or different levels of care.

A total of 8128 patients (median number of participants per study: 125, 17–3693) were included and 6345 were exposed to the ICP. Of these, mean ages ranged from 42 to 75.9±7.4 years. The interventions had been developed locally for a range of purposes, in either hospital secondary or tertiary settings: improving service coordination, increasing service efficiency, supporting practice change, improving patient outcomes, ensuring adherence to best practice guidelines. Most had been implemented in order to achieve multiple aims (Table 2). The ICPs were considered a complex intervention [47–49], as they comprise a number of separate essential elements. None of the studies included in the review were underpinned by explicit theories of ICPs' active ingredients or their generative effects. Moreover, the information provided on ICP development and implementation processes was varied and in no case was any evidence

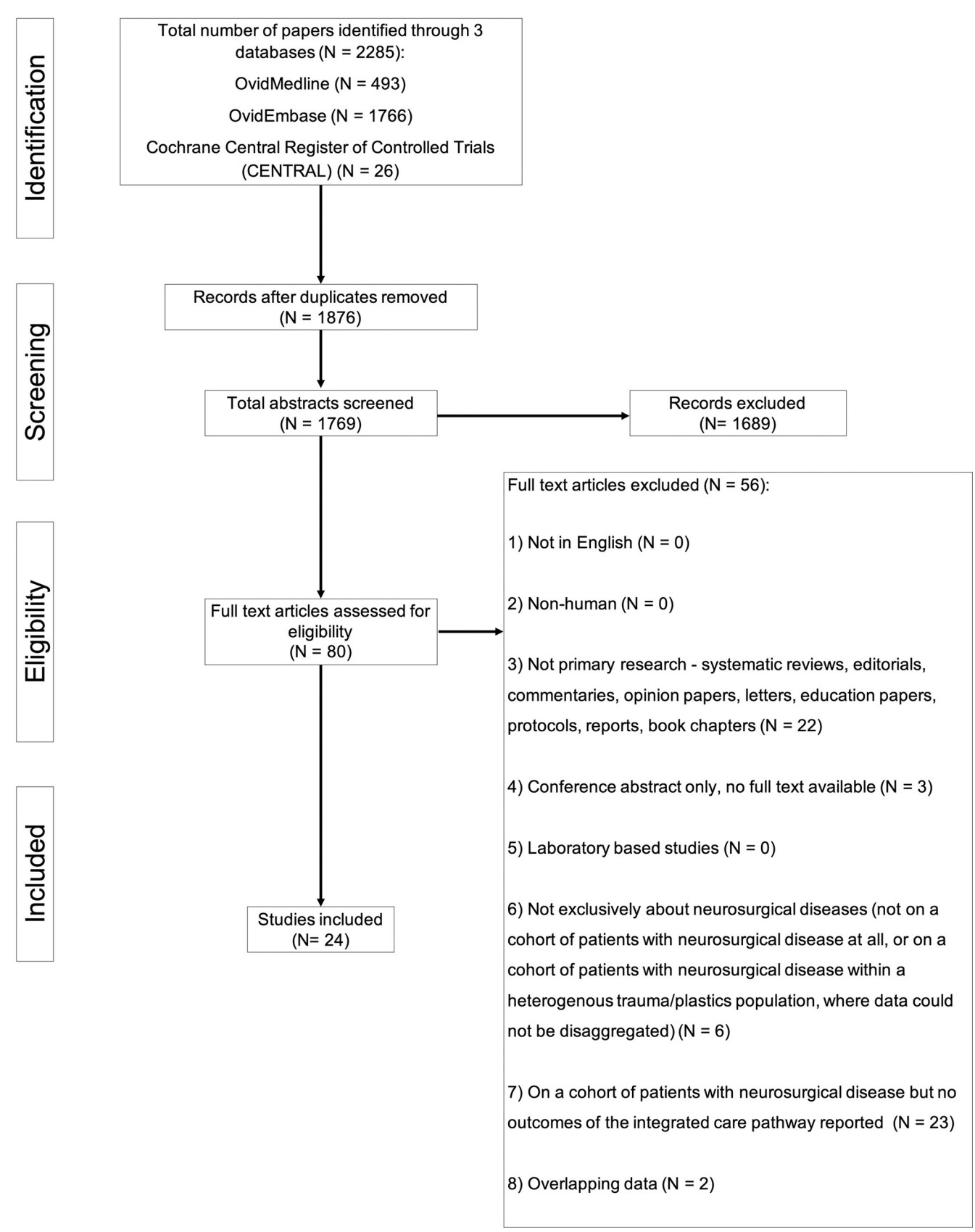

**Fig 1. PRISMA flow diagram of included and excluded studies.**

**Table 1. Study characteristics of the included primary studies.**

| Study | Country | Setting | Study design | Neurosurgery classification | Aim of the ICP | Outcomes | Sample size (n) | Gender (% male) | Age (mean ±SD) |
|---|---|---|---|---|---|---|---|---|---|
| Adogwa 2018 | USA | Hospital, tertiary care | Cohort | Spine | To determine if neurosurgery geriatric co-management reduces ICU admission rates after spine surgery | ICU admission rate after surgery. Postoperative complications | 125 | Intervention: 41% male; control: 36% male | Intervention: 73.6±6.0; control 73.0 ±4.9 |
| Akhunbay-Fudge 2019 | UK | Hospital, secondary care | Case series | CSF diversion | To collect digital retinal photographs as facilitate clinical assessment of shunt malfunction | Clinical decisions made | 67 | NA | NA |
| Akins 2019 | USA | Hospital, tertiary care | Case series | Miscellaneous | To provide more consistent care and to improve communication, to improve outcomes and care efficiency | Hospital related complications, functional outcomes, discharge destinations, and avoidable delays in care | 3693 | NA | NA |
| Aldana 2010 | USA | Hospital, secondary/ tertiary care | Case series | Paediatric | To better manage children with spinal defects | Knowledge of child's medical condition, care plans, need for medical and prosthetic devices, medical-social needs, reduction in physician and allied health care contacts, transportation costs | 139 | NA | NA |
| Allali 2017 | Switzerland | Hospital, secondary care | Case series | CSF diversion | To better identify older adults with iNPH from its mimics and to allow better management | Gait parameters, cognition | 125 | 65.6% male | 75.9±7.4 |
| Bapat 2017 | UK | Hospital, secondary care | Cohort | Neurovascular | To improve quality of care for patients presenting with chronic subdural haematoma | Use of anti-coagulant or anti-platelet agents, timing of surgery, complications, morbidity and mortality, recurrence, LOS and destination at discharge | 121 (intervention: 68; control: 53) | Intervention: 70.6% male; control: 71.7% male | Intervention: median 74 (36–91); control: median 74 (36–91) |
| Bohl 2017 | USA | Hospital, secondary care | Cohort | Skull base | To reduce 30-d readmissions due to delayed hyponatremia following transsphenoidal surgery | Postoperative LOS, postoperative inpatient sodium levels, and need for preoperative or postoperative hydrocortisone | 417 (intervention: 188; control: 229) | Intervention: 56.9% male; control 49.8% male | Intervention: 51.9±16.3; control: 52.5 ±16.9 |
| Brown 2018 | USA | Hospital, tertiary care | Cohort | Miscellaneous | To screen for the risk and presence of delirium, and to implement non-pharmacologic interventions to those patients at high risk of developing or have developed delirium | Change in incidence of hospital-acquired delirium, delirium duration, overall LOS, restraint use, sitter use, disposition to nursing facility, and 30-day readmission rate. | 1501 (intervention: 749; control: 752) | Intervention: 49% male; control 47% male | Intervention: 67.1±11.2; control: 67.1 ±11.1 |

*(Continued)*

**Table 1.** (Continued)

| Study | Country | Setting | Study design | Neurosurgery classification | Aim of the ICP | Outcomes | Sample size (n) | Gender (% male) | Age (mean ±SD) |
|-------|---------|---------|--------------|----------------------------|----------------|----------|-----------------|-----------------|----------------|
| Buell 2019 | UK | Hospital, secondary/ tertiary care | Case series | Spine | To reduce the time of presentation to diagnosis or exclusion of CES. | Time interval between the patient's arrival to the ED and MRI preliminary report. | 17 | NA | NA |
| Carminucci 2016 | USA | Hospital, secondary/ tertiary care | Cohort | Skull base | To optimise postoperative care of transsphenoidal surgery | Neurosurgical and endocrine complications, LOS, and rates of hospital readmission and unscheduled clinical visits. | 214 (intervention: 101; control: 113) | Intervention: 51%; control: 49% | Intervention: 52.4±1.4; control: 50.7 ±1.4 |
| Chern 2010 | USA | Hospital, secondary/ tertiary care | Cohort | CSF diversion | To expedite care of patients w CSF shunt malfunction | ED process measures (timeliness), clinical outcomes (admission rate, shunt surgery rate, and LOS) | 245 (intervention: 113; control: 132) | NA | NA |
| Chung 2005 | Korea | Hospital, tertiary care | Cohort | Spine | To improve LOS and hospital costs in patients undergoing lumbar surgery | LOS and cost. | 119 (intervention: 58; control: 61) | Intervention: 56.9% male; control: 62.3% male | Intervention: 49.7±16.7; control: 51.3 ±15.4 |
| Cohen 2007 | USA | Hospital, secondary/ tertiary care | Case series | Functional | To determine that rehabilitation following DBS improves outcomes | FIM, UPDRS scores and levodopa dosage. | 73 | 68.5% male | 60.6 |
| Debono 2017 | France | Hospital, secondary care | Case series | Spine | To determine if dedicated fast-tracking outpatient lumbar microdiscectomy, could achieve patient satisfaction, raises complications, and return to normal ADL | Patient satisfaction, complications, and return to normal ADL | 201 | 71.1% male | 42 |
| Giorgi 2020 | Italy | Hospital, secondary/ tertiary care | Case series | Spine | To determine if organisational protocol for emergency spinal surgery reduces time from admission to surgery | Time duration from admission to surgery | 19 | 57.9% male | 49.9 |
| Jin 2008 | Netherlands | Hospital, secondary/ tertiary care | Cohort | Trauma | To reduce time for complete workup for severely, and multiply injured patients, and to improve functional outcomes and mortality rates | TBI-related mortality and functional neurological outcome | 108 (intervention: 49; control: 59) | Intervention: 69% male; control: 61% male | Intervention: 49; control: 44 |
| Kurlander 2020 | USA | Hospital, secondary/ tertiary care | Cohort | Paediatric | To reduce or eliminate blood transfusion in patients undergoing open surgery for craniosynostosis. | Estimated blood loss, transfusion rate, and intraoperative transfusion | 41 | NA | NA |

(*Continued*)

**Table 1.** (Continued)

| Study | Country | Setting | Study design | Neurosurgery classification | Aim of the ICP | Outcomes | Sample size (n) | Gender (% male) | Age (mean ±SD) |
|---|---|---|---|---|---|---|---|---|---|
| Namiranian 2018 | USA | Hospital, tertiary care | Cohort | Spine | To determine if a multidisciplinary spine board was concurrent with an overall decrease in the utilization of lumbar spine surgeries for elective cases of low back pain | Surgery duration, estimated blood loss, packed red blood cell transfusion, destination after surgery, and LOS in hospital or ICU, surgical complications | 152 (intervention: 51; control: 101) | | |
| Playford 2002 | UK | Hospital, secondary care | Case series | Spine | To assess the rates of goal achievement and the sources of variance, in a inpatient rehabilitation protocol following spinal lesion | The numbers and categories of goals and the rates of goal achievement, variance patterns | 85 | NA | NA |
| Pritchard 2004 | UK | Hospital, secondary care | Cohort | Neurovascular | To reduce dysfunctional psychosocial stress following aneurysmal subarachnoid haemorrhage | Cost-effectiveness | 326 (intervention: 184; control: 142) | Intervention: 39% male; control: 37% male | NA |
| Scanlon 2004 | USA | Hospital, secondary care | Case series | Spine | To determine if outpatient laminectomy programme is feasible, based on patient satisfaction | LOS in the PACU, level of pain on discharge, return to the hospital within 24 hours, patient satisfaction score | 27 | NA | NA |
| Sethi 2017 | USA | Hospital, tertiary care | Cohort | Spine | To minimise perioperative risk and maximise QOL in adult scoliosis surgery | Operative time, number of levels fused, and LOS. Surgical complications within 30 days or up to 1 year. | 140 (intervention: 69; control: 71) | Intervention: 16% male; control: 35% male | Intervention: mean 65.5 ±10.5; control: 62.0±13.4 |
| Soffin 2019 | USA | Hospital, secondary/ tertiary care | Case series | Spine | To implement ERAS patient care pathway for ACDF patients CDA—improve LOS and outcome | LOS and reasons for LOS exceeding 23 hours, pathway compliance, prevalence of opioid tolerance at baseline, and the effect of opioid tolerance on outcomes. | 33 | 45.5% male | NA |
| Wang 2019 | China | Hospital, secondary/ tertiary care | Cohort | Neuro oncology | To implement ERAS protocol for elective craniotomies— improve periop care and outcome | LOS, 30d readmission rates, postoperative morbidity, surgical and non-surgical complications, functional recovery status and patient satisfaction ratings. | 140 (intervention: 70; control: 70) | Intervention: 31% male; control: 37% male | Intervention: median 51 (19–67); control: median 49 (18–65) |

ADL = activities of daily living; ED = emergency department FIM = Functional independence measure; ICU = intensive care unit; LOS = length of stay; MRI = magnetic resonance imaging NA = not available; PACU = postanaesthesia care unit UPDRS = Unified Parkinson Disease Rating Scale

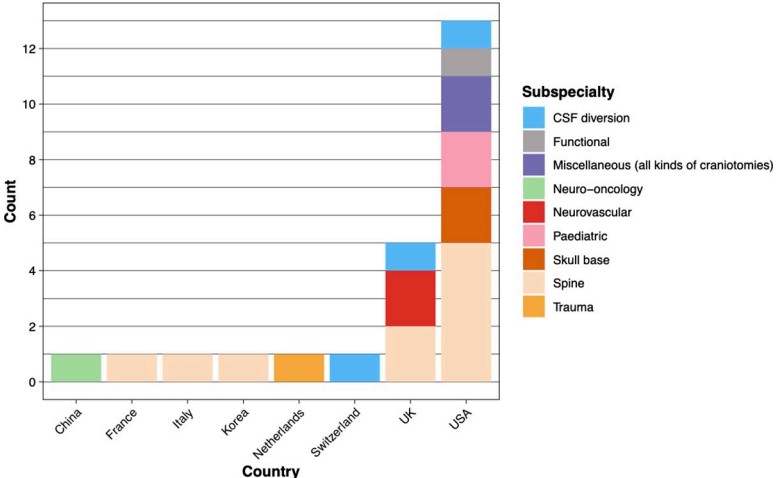

**Fig 2. Country of origin and neurosurgical specialities of the included studies.**

provided for the selection of any ICP component of the intervention to be assessed. In several cases it was possible to make inferences about authors' implicit assumptions. Only 10 of the 25 ICPs were presented in detail as a flow diagram figure. The interventions described by the studies in the review varied in terms of their key components which we summarise in Table 3, using the definition developed by the European Pathway Association (EPA) [6]. Hence, we were also unable to meet our objectives of defining 'active ingredients' (setting, context, and population) for ICPs to be successful or factors contributing to the ineffectiveness of ICPs.

## Outcome measures

We identified an extensive range of outcomes (Table 2). These outcome measures were considered and categorised into three main areas: those relating to the patient, those relating to personnel working experience and finally those relating to system.

The most frequently measured patient outcomes were complications (n = 9), readmission rates (n = 5), discharge destinations (n = 5) need for medication/devices/social services

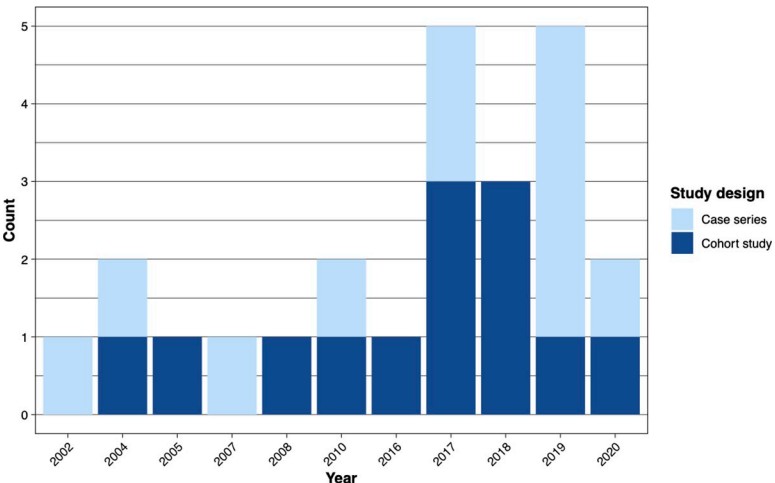

**Fig 3. Trend of the year of ICP-related neurosurgical publications.**

**Table 2. Included studies fit with the core components of the EPA definition of an ICP.**

| Study | Neurosurgery classification | An explicit statement of goals and key elements of care based on evidence and/or best practice | Facilitation of communication | Coordination of roles and sequencing of activities of the multidisciplinary team | The facilitation of communication with patients and their relatives (Blank) | Forms part or all of the patients documentation | Includes monitoring, and evaluation of variances and outcomes | The identification of appropriate resources (Blank) |
|---|---|---|---|---|---|---|---|---|
| | | | | | Core components of the ICP | | | |
| Adogwa 2018 | Spine | ✓ | | ✓ | | | | |
| Akhunbay-Fudge 2019 | CSF diversion | ✓ | ✓ | ✓ | | ✓ | ✓ | ✓ |
| Akins 2019 | Miscellaenous | ✓ | ✓ | ✓ | ✓ | ✓ | ✓ | |
| Aldana 2010 | Pediatric | ✓ | ✓ | ✓ | ✓ | | | |
| Allali 2017 | CSF diversion | ✓ | ✓ | ✓ | | ✓ | ✓ | ✓ |
| Bapat 2017 | Neurovascular | ✓ | ✓ | ✓ | | ✓ | | |
| Bohl 2017 | Skull base | ✓ | ✓ | ✓ | | ✓ | ✓ | |
| Brown 2018 | Miscellaenous | ✓ | ✓ | ✓ | | ✓ | | |
| Buell 2019 | Spine | ✓ | ✓ | ✓ | | ✓ | | |
| Carminucci 2016 | Skull base | ✓ | ✓ | ✓ | | ✓ | ✓ | |
| Chern 2010 | CSF diversion | ✓ | ✓ | ✓ | | ✓ | ✓ | |
| Chung 2005 | Spine | ✓ | ✓ | ✓ | | ✓ | ✓ | |
| Cohen 2007 | Functional | ✓ | ✓ | ✓ | | ✓ | ✓ | |
| Debono 2017 | Spine | ✓ | ✓ | ✓ | | ✓ | | |
| Giorgi 2020 | Spine | ✓ | ✓ | | | | | |
| Jin 2008 | Trauma | ✓ | ✓ | | | ✓ | | |
| Kurlander 2020 | Pediatric | ✓ | ✓ | ✓ | | ✓ | | |
| Namiranian 2018 | Spine | ✓ | ✓ | ✓ | | ✓ | | |
| Playford 2002 | Spine | ✓ | | | | ✓ | ✓ | |
| Pritchard 2004 | Neurovascular | ✓ | ✓ | ✓ | ✓ | ✓ | ✓ | |
| Scanlon 2004 | Spine | ✓ | ✓ | ✓ | ✓ | | | |
| Sethi 2017 | Spine | ✓ | ✓ | ✓ | | ✓ | ✓ | |
| Soffin 2019 | Spine | ✓ | ✓ | | | | | |
| Wang 2019 | Neuro oncology | ✓ | | ✓ | | ✓ | ✓ | ✓ |

(n = 4), mortality (n = 3), patient satisfaction (n = 3), functional outcomes (n = 3), return to independence (n = 3), duration of surgery (n = 3). Patient satisfaction was measured subjectively via phone survey (n = 3). Other reported patient outcomes included need for repeat surgery (n = 2), morbidity (n = 2) and patient/family education (n = 1). Professional outcomes such as team satisfaction and communication were not reported in the included studies. Length of stay (LOS) (n = 11) was the most commonly used indicator for system level outcomes. Other system level indicators reported were timeliness/avoidable delays to care or assessment (n = 4), costs (n = 3), and finally pathway compliance and variance (n = 2)

**Table 3. ICP checklist regarding implementation, reporting and delivery.**

| Point to be reviewed | Tickbox | |
|---|---|---|
| Define aim/problem to address and set objectives (goals) in the beginning. | Yes | No |
| State field or subspecialty (e.g. neurovascular surgery, tumour surgery, spinal surgery etc.) | Yes | No |
| Define intervention and control group with clear inclusion and exclusion criteria stated | Yes | No |
| Define areas of improvement (may include more than one) e.g. clinical outcomes, facilitation of communication (patient-clinician or clinician-family, or both), cost-savings, educational, etc. | Yes | No |
| Define element of patient care pathway that is addressed e.g. pre-operative, post-operative, full patient journey, diagnostic, follow up, etc. | Yes | No |
| Details of the process of ICP development and implementation maturity e.g pilot, under review/ investigation, implemented etc. | Yes | No |
| Define choice of evidence in use to support decision making (best practice, best evidence, expert advise, etc) | Yes | No |
| State roles of members involved in ICP | Yes | No |
| e.g nurse practitioner coordinating part of patient journey, specific review of specialties (complex geriatric assessment of elderly), allied health professional roles in rehabilitation pathways etc.) | | |
| State resources needed e.g. financial, time, human e.g coordinator roles, additional staffing etc. | Yes | No |
| Define and report outcomes with follow up, and further re-evaluation of service | Yes | No |
| Standardised reporting of demographics and results with included key ingredients as per Allen et al 2009. | Yes | No |
| e.g. implemented over a specified time frame; activities specified by professional role; decision support aide included; formed part of the patient record; based on best evidence or best practice; variance tracking; locally developed and implemented; supporting education and training initiatives etc. | | |

## Subspecialties

**CSF diversion.**   Three studies reported the function of ICP related to CSF flow pathologies, although relating to different parts of the patient journey [28, 36, 39]. Two were case series without a control group [36, 39], and one was a cohort study [28].

Two studies used ICPs in the setting of suspected shunt malformation. Akhunbay-Fudge et al. evaluated the use of an assessment pathway utilising a digital retinal camera system to assess for papilloedema remotely but reported no outcome measures [36]. Chern et al. designed a cohort study to evaluate the fast track preoperative protocol where eligible patients at risk of shunt failures entered the ICP for further workup [28]. The ICP was compared with preprotocol periods as control.

Outcome metrics to evaluate its effectiveness included admission rate, LOS, need for repeat shunt surgery, and timeliness.

Allali et al. determined the feasibility of a protocol using cognitive and gait quantification to identify normal pressure hydrocephalus in elderly patients, distinguishing it from its mimics such as Parkinson's disease or vascular dementia [39].

**Functional.**   The use of ICP in functional neurosurgery was reported in one case series.

Cohen et al. evaluated the effectiveness of a multidisciplinary rehabilitation model for Parkinson's disease patients who had undergone DBS. Outcome was assessed using 'return to independence' as judged by the Functional Independence Measure (FIM) and Unified Parkinson's Disease Rating Scale (UPDRS) scores [41].

**Neuro-oncology.**   Wang et al. established a neurosurgical enhanced recovery after surgery (ERAS) programme in a Chinese tertiary care medical centre, for patients undergoing elective craniotomy for primary brain tumours [35]. This ERAS protocol appeared to have significant benefits over its comparator–conventional perioperative management. Outcome measures

were adverse events (30 day readmission rates, both surgical and non-surgical complications, LOS, morbidity, mortality), functional recovery and patient satisfaction.

**Neurovascular.**   Two cohort studies reported the function of ICP related to neurovascular surgery [24, 33]. Bapat and colleagues, together with a multidisciplinary team of neurosurgeons, neuroanaesthetists and rehabilitation therapists developed a ICP for elderly patients with chronic subdural haemorrhage (CSDH), to enhance preoperative optimisation and reduce time to surgery [24]. Outcome metric assessed and reported were adverse events (LOS, complications, mortality, recurrence), discharge destinations, timing of surgery, use of anticoagulant of antiplatelet agents.

Pritchard et al. assessed the cost effectiveness of an enhanced Specialist Liaison Nurse (SLN) service which sought to reduce dysfunctional psychosocial stress in sufferers of aneurysmal subarachnoid haemorrhage (aSAH) [33]. The only outcome measure was cost effectiveness.

**Paediatric.**   The use of ICP function neurosurgery was reported in one case series,[4] and one cohort study [31].

Aldana and colleagues set up a comprehensive multidisciplinary clinic to better assess spinal defects such as meningocoele, myelocystocoele spina bifida occulta syringomyelia amongst many others [38]. Outcome measures were parents' knowledge of child's medical condition, care plans, need for medical and prosthetic devices, reduction in physician and allied health care contacts, and transportation costs.

Kurlander et al. performed a cohort study to assess quality improvement blood conservation protocol for craniosynostosis [31]. It resulted in a 66% transfusion-free rate at time of discharge compared to 0% in the group without any conservation protocol.

**Skull base.**   Two cohort studies reported the function of ICP related to postoperative management following transsphenoidal surgery for sellar lesions [25, 27].

Common outcome measures reported in these skull base studies were readmission, surgical and endocrinological complications, LOS, postoperative inpatient sodium levels, and need for preoperative or postoperative hydrocortisone.

**Spine.**   Ten studies reported the function of ICP related to spinal surgeries for lumbar pathologies or adult scoliosis [23, 29, 32, 34, 40, 42–46]. Four involved a control group whereas six were case series. The specific interventions regarding the pathways described in the included studies showed considerable variation. The studies mainly focused on ICPs for surgical care or perioperative phase in order to guide surgical management and reduce its delay, whilst one study investigated pain management.

Common outcome measures reported complications, ICU admission, delays to assessment MRI report, duration of surgery, estimated blood loss, LOS, costs, return to dependence and ADL, patient satisfaction (assessed by phone survey), destination after discharge, ICP compliance and variance pattern.

**Trauma.**   Jin et al. introduced a streamlined workflow concept that included direct computed tomography (CT) scanning in the trauma room in patients with severe traumatic brain injury (TBI) [30]. The cohort study measured TBI related mortality and functional outcomes.

## Discussion

### Summary of findings

To the best of our knowledge, this is the first systematic review to identify and assess ICPs for neurosurgical diseases. Twenty-four original articles were identified across a range of neurosurgical pathologies and settings. All ICP were based on a single centre experience and 13 of 24 compared practice before and after adoption, to evaluate added benefit. Few studies utilised

patient perspective in their evaluation, preferring often isolated performance metrics such as length of hospitalisation. Approaches to the design and iteration of ICPs upon implementation were not reported.

## ICPs can work in neurosurgical practice

ICPs are considered a road map for care, which rely on multi-disciplinary involvement such as doctors, nurses, physiotherapists and occupational therapists, and other healthcare professionals. ICPs aim to improve quality and efficiency. They have been adopted in a variety of health care specialties and settings often with positive results, as seen in orthopaedics [3–5, 8].

ICPs are therefore likely to benefit Neurosurgery given its requirements for multi-disciplinary cooperation both within a tertiary centre but also across a region. The examples identified in this review indicate their relevance and value across sub-specialities of Neurosurgery, for example to ensure patients receive relevant clinical assessments or interventions in a timely and efficient fashion, to reduce variation in practice or readmissions, and improve length of stay and patient satisfaction. Reassuringly this review also demonstrate clinicians are increasingly engaged in initiatives to improve the delivery of care through the redesign of existing services [3–5, 50], including the use of ICPs as seen in Fig 3.

However, there were notable omissions both in the design and evaluation of identified studies. For example, the identified ICP focus on service delivery by the tertiary centre, and do not incorporate regional care pathways, which will be relevant in the delivery of an emergency tertiary service for example. Furthermore, there is an underrepresentation of the patient voice in these included studies. We identified surprisingly little evidence regarding the impact of ICPs on patient experiences of services, beyond measures of reported patient satisfaction. Measurement of patient satisfaction were limited only to phone surveys which may be highly subjected to detection bias. Outcome measures such as LOS may be effective surrogates, but any such use should follow their validation.

## Future neurosurgical ICPs should consider the growing methodological literature around their design and evaluation

As highly complex interventions, ICP challenge linkage of particular elements of initiatives to effects [47–49], and these aforementioned omissions limit the thorough evaluation of reported ICPs, and potentially their wider adoption.

Process evaluation of ICPs is complex and requires a combination of quantitative and qualitative methods to inform policy and practice [51]. Whilst RCTs are considered a gold-standard, their delivery in this setting is more difficult as interventions are often multifaceted and harder to separate between arms [52]. Variants including cluster randomised or step-wedge trials are potential alternatives [53]. The UK Medical Research Council guidance [47–49], outlines that evaluation requires good working relationships with all stakeholders involved in ICP development. Problems that are identified during implementation can be adjusted, as per a quality improvement process, but not at the point at which the ICP is being evaluated [54]. Active correction is therefore more appropriate at the development or feasibility trial stage [55]. The MRC recommendations also include the development and evaluation of for complex interventions through iterative phases, to ensure the relevance of each intervention [47–49]. An ICP should therefore be a work in progress that can be further improved on through repeated quality improvement cycles [56]. The use of pre-specified targets and timepoints for evaluation, including a control group for comparison, may allow a team to know if the project goals have been achieved and consider what interventions to retain, improve, or discard in future cycles of ICP development [57–59].

Reflecting on the data omissions in this review and the implications for study interpretation, a reporting framework would be of benefit [60, 61]. Table 3 outlines a proposed checklist for Neurosurgical ICPs regarding its implementation, reporting and delivery based on wider experience and findings in this review [16, 17]. This checklist is intentionally generic, representing a minimum set of critically important outcomes to report in all studies evaluating the introduction and evaluation of ICPs and should not restrict investigators in their reporting of additional relevant outcomes. In future, this could be further refined by a Delphi consensus of various stakeholders–neurosurgeons, radiologists, oncologists, nurses, allied healthcare professionals, health-economists.

## Strengths and limitations of the review

This review employed a pre-specified, registered protocol and variations to the protocol have been explicitly stated. The literature search was comprehensive, identifying relevant studies from three databases, and the reporting of this review follows PRISMA guidelines [14].

Limitations of this review are that we were only able to include publications written in English, due to resource constraints. However, international publications were included which may reduce selection bias. We also acknowledge a potential issue of publication bias, with studies reporting fewer positive outcomes almost certainly underrepresented in the review. Further well established ICPs such as the metastatic cord compression and head injury pathways, published by the UK National Institute for Health and Care Excellence (NICE) were not identified using the literature database searches [62, 63]. NICE pathways are not indexed on the academic databases we searched as their recommendations are generally formed using expert consensus based on available evidence, and adoption or effectiveness not routinely evaluated in scientific papers. An awareness that national ICPs exist supports but would not have changed the result of this SR. We highlight the challenges inherent when defining models of integrated care, given the lack of agreed definition and clear boundaries to the term. This limitation may have resulted relevant work being excluded from this review. During the selection of studies, it was particularly challenging to discern between new models of care that are 'integrated' from those that are not, as numerous terms were used interchangeably to describe the management of clinical care processes within the literature. However, rigorous and blinded screening, together with consensus discussion helps to mitigate this issue.

## Conclusion

ICPs in Neurosurgery have been developed and may have a beneficial role in neurosurgical care. However, examples so far are limited to single institutions, have an uncertain development process and longer-term legacy, whilst appear to lack patient perspective both in design and evaluation. This limits firm conclusions on its effectiveness. Moreover, evaluation has used an audit change cycle, precluding evaluation of single measures (if complex interventions) and open to performance bias. Experiences from parallel fields, suggest these areas must be overcome, to ensure a generalisable and sustainable ICP. Their development and generalisation would benefit from a reporting framework and accordingly, a checklist for ICPs regarding its implementation, reporting and delivery has also been proposed.

## Supporting information

**S1 Checklist. PRISMA checklist.**
(PDF)

**S1 Table. Full search phrases used for the three respective databases.**
(DOCX)

**S2 Table. Inclusion and exclusion criteria used to select studies for the review.**
(DOCX)

**S3 Table. Joanna Briggs Institute quality assessment checklist for cohort studies.**
(DOCX)

**S4 Table. Joanna Briggs Institute quality assessment checklist for case series.**
(DOCX)

## Author Contributions

**Conceptualization:** Keng Siang Lee, Benjamin Davies.

**Data curation:** Keng Siang Lee, Stefan Yordanov.

**Formal analysis:** Keng Siang Lee.

**Investigation:** Keng Siang Lee.

**Methodology:** Keng Siang Lee, Stefan Yordanov, Daniel Stubbs.

**Project administration:** Keng Siang Lee.

**Software:** Keng Siang Lee.

**Supervision:** Benjamin Davies.

**Visualization:** Keng Siang Lee, Benjamin Davies.

**Writing – original draft:** Keng Siang Lee.

**Writing – review & editing:** Keng Siang Lee, Stefan Yordanov, Daniel Stubbs, Ellie Edlmann, Alexis Joannides, Benjamin Davies.

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
