## [Decision Letter · Decision Letter 0]

1 Jul 2021

PONE-D-21-08319

Integrated care pathways in Neurosurgery: A systematic review

PLOS ONE

Dear Dr. Keng Siang Lee

Thank you for submitting your manuscript to PLOS ONE. After careful consideration, we feel that it has merit but does not fully meet PLOS ONE’s publication criteria as it currently stands. Therefore, we invite you to submit a revised version of the manuscript that addresses the points raised during the review process.

I would appreciate if you pay a careful attention in your response to the reviewer's comments.

We look forward to receiving your revised manuscript.

Kind regards,

Ehab Farag, MD FRCA FASA

Academic Editor

PLOS ONE

Journal Requirements:

Reviewers' comments:

Reviewer's Responses to Questions

**Comments to the Author**

1. Is the manuscript technically sound, and do the data support the conclusions?

Reviewer #1: Partly

2. Has the statistical analysis been performed appropriately and rigorously? 

Reviewer #1: I Don't Know

3. Have the authors made all data underlying the findings in their manuscript fully available?

Reviewer #1: Yes

4. Is the manuscript presented in an intelligible fashion and written in standard English?

Reviewer #1: Yes

5. Review Comments to the Author

Reviewer #1: The manuscript does not discuss what it was meant to find out possibly due to lack of data. The discussion is also vague and extensive. Clearcut the goals and outcome with a focussed discussion would be better.

6. PLOS authors have the option to publish the peer review history of their article (what does this mean?). If published, this will include your full peer review and any attached files.

Reviewer #1: No

---

## [Author Response · Author response to Decision Letter 0]

15 Jul 2021

Professor Ehab Farag, MD FRCA FASA 

Academic Editor 

PLOS ONE

15 July 2021

Dear Professor Ehab Farag, 

Re. Integrated care pathways in Neurosurgery: A systematic review [PONE-D-21-08319]

We thank you for the editorial and reviewers’ comments, and for providing us the opportunity to revise our manuscript. We appreciate the careful review and constructive suggestions from all reviewers and have responded to the comments below. 

Following this letter are the review comments with our response in italics, including how and where the text was modified. Enclosed are two copies of the edited manuscript, one with the changes highlighted using the Microsoft Word Tracker system and a second copy with the changes in normal typeface. The revision has been developed in consultation with all co-authors, and each author has given approval to the final form of this revision.

Keng Siang Lee and Benjamin Davies

Bristol Medical School, University of Bristol, Bristol, UK. Department of Clinical Neurosurgery, University of Cambridge, Cambridge, UK

Email: kl17746@bristol.ac.uk (KSL)

Email: bd375@cam.ac.uk (BD)

 

Journal Requirements:

Response: 

Thank you for pointing this out. We have now formatted the revised manuscript accordingly to the journal’s requirement.

 

Comments to the Author

1. Is the manuscript technically sound, and do the data support the conclusions?  The manuscript must describe a technically sound piece of scientific research with data that supports the conclusions. Experiments must have been conducted rigorously, with appropriate controls, replication, and sample sizes. The conclusions must be drawn appropriately based on the data presented.

Reviewer #1: Partly

Response: 

Thank you. We have responded further to your additional comments below.

2. Has the statistical analysis been performed appropriately and rigorously?

Reviewer #1: I Don't Know

Response: 

Thank you. Given the heterogeneity in data reporting, formal meta-analysis of the included studies was not possible. Therefore, we felt it was more appropriate to report the data narratively, with descriptive statistics only. This has, we believe, enabled a comprehensive overview of studies developing ICPs for Neurosurgery, including a recognition of potential limitations that should inform their development in the future.

3. Have the authors made all data underlying the findings in their manuscript fully available? The PLOS Data policy requires authors to make all data underlying the findings described in their manuscript fully available without restriction, with rare exception (please refer to the Data Availability Statement in the manuscript PDF file). The data should be provided as part of the manuscript or its supporting information, or deposited to a public repository. For example, in addition to summary statistics, the data points behind means, medians and variance measures should be available. If there are restrictions on publicly sharing data—e.g. participant privacy or use of data from a third party—those must be specified.

Reviewer #1: Yes

Response: 

Thank you.

4. Is the manuscript presented in an intelligible fashion and written in standard English? PLOS ONE does not copyedit accepted manuscripts, so the language in submitted articles must be clear, correct, and unambiguous. Any typographical or grammatical errors should be corrected at revision, so please note any specific errors here.

Reviewer #1: Yes

Response: 

Thank you.

5. Review Comments to the Author  Please use the space provided to explain your answers to the questions above. You may also include additional comments for the author, including concerns about dual publication, research ethics, or publication ethics. (Please upload your review as an attachment if it exceeds 20,000 characters)

Reviewer #1: The manuscript does not discuss what it was meant to find out possibly due to lack of data. The discussion is also vague and extensive. Clearcut the goals and outcome with a focussed discussion would be better.

Response: Thank you – despite a comprehensive search strategy, there is somewhat limited published examples of ICP in surgery

Whilst we think these findings support value in their role, across different neurosurgical disciplines, our detailed analysis of their development, design and evaluation identified weaknesses compared to best practice in ICP development.

Due to the heterogeneity in the data, only a narrative synthesis was possible which has created a lengthy article. However, overall we feel that this is a comprehensive representation of the field currently, and therefore a useful resource for others to build on. 

We have made various amendments to the text to highlight the aims, and clarify the message. These changes have been made particularly in the subsection ‘Future neurosurgical ICPs should consider the growing methodological literature around their design and evaluation’. We hope these now meet your approval.

6. PLOS authors have the option to publish the peer review history of their article (what does this mean?). If published, this will include your full peer review and any attached files.   Do you want your identity to be public for this peer review? For information about this choice, including consent withdrawal, please see our Privacy Policy.

Reviewer #1: No

Response: 

Thank you for taking time to review our manuscript and providing constructive feedback which could only improve our work.

---

## [Editor Report · Decision Letter 1]

21 Jul 2021

Integrated care pathways in Neurosurgery: A systematic review

PONE-D-21-08319R1

Dear Dr. Keng Siang Lee

We’re pleased to inform you that your manuscript has been judged scientifically suitable for publication and will be formally accepted for publication once it meets all outstanding technical requirements.

Kind regards,

Ehab Farag, MD FRCA FASA

Academic Editor

PLOS ONE
---

## [Editor Report · Acceptance letter]

23 Jul 2021

PONE-D-21-08319R1 

Integrated care pathways in Neurosurgery: A systematic review 

Dear Dr. Lee:

I'm pleased to inform you that your manuscript has been deemed suitable for publication in PLOS ONE. Congratulations! Your manuscript is now with our production department. 

Kind regards, 

on behalf of

Dr. Ehab Farag 

Academic Editor

PLOS ONE